# Involvement of a Multidrug Efflux Pump and Alterations in Cell Surface Structure in the Synergistic Antifungal Activity of Nagilactone E and Anethole against Budding Yeast *Saccharomyces cerevisiae*

**DOI:** 10.3390/antibiotics10050537

**Published:** 2021-05-06

**Authors:** Yuki Ueda, Yuhei O. Tahara, Makoto Miyata, Akira Ogita, Yoshihiro Yamaguchi, Toshio Tanaka, Ken-ichi Fujita

**Affiliations:** 1Graduate School of Science, Osaka City University, Sumiyoshi-ku, Osaka 558-8585, Japan; g38dw2nop@gmail.com (Y.U.); taharayuhei@osaka-cu.ac.jp (Y.O.T.); miyata@osaka-cu.ac.jp (M.M.); ogita@osaka-cu.ac.jp (A.O.); yyamaguchi@osaka-cu.ac.jp (Y.Y.); seitaiteibunshi_labo@yahoo.co.jp (T.T.); 2Research Center for Urban Health and Sports, Osaka City University, Sumiyoshi-ku, Osaka 558-8585, Japan

**Keywords:** antifungal, nagilactone E, drug resistance, multidrug efflux pump, quick-freeze deep-etch electron microscopy

## Abstract

Nagilactone E, an antifungal agent derived from the root bark of *Podocarpus nagi*, inhibits 1,3-β glucan synthesis; however, its inhibitory activity is weak. Anethole, the principal component of anise oil, enhances the antifungal activity of nagilactone E. We aimed to determine the combinatorial effect and underlying mechanisms of action of nagilactone E and anethole against the budding yeast *Saccharomyces cerevisiae*. Analyses using gene-deficient strains showed that the multidrug efflux pump *PDR5* is associated with nagilactone E resistance; its transcription was gradually restricted in cells treated with the drug combination for a prolonged duration but not in nagilactone-E-treated cells. Green-fluorescent-protein-tagged Pdr5p was intensively expressed and localized on the plasma membrane of nagilactone-E-treated cells but not in drug-combination-treated cells. Quick-freeze deep-etch electron microscopy revealed the smoothening of intertwined fiber structures on the cell surface of drug-combination-treated cells and spheroplasts, indicating a decline in cell wall components and loss of cell wall strength. Anethole enhanced the antifungal activity of nagilactone E by enabling its retention within cells, thereby accelerating cell wall damage. The combination of nagilactone E and anethole can be employed in clinical settings as an antifungal, as well as a food preservative to restrict food spoilage.

## 1. Introduction

Antibiotics and preservatives used in food items should have minimal side effects in humans and specifically target microbes. Fungi, including molds and yeasts, are eukaryotes and have metabolic processes and cell structures similar to those of humans, which makes the development of targeted treatment options against fungal pathogens a challenge. In addition, the emergence of drug-resistant fungi in recent years has become a serious problem for effective chemotherapy against deep-seated mycoses [1]. Therefore, drug-resistant fungi need to be controlled using new drugs or combination therapy. In combination therapy, resistance of fungal cells to a drug can be overcome using another drug that suppresses this resistance.

Nagilactones are norditerpene dilactones isolated from the root bark of *Podocarpus nagi* [2]. Nagilactone C, D, and F have insect feeding deterrent activity against *Lactuca sativa* [3]. Nagilactone C has insecticidal activity against *Musca domestica* Linnaeus [4] and potent antitumor activity against Yoshida sarcoma [5,6]. Nagilactone D and E (nagilactone E; Figure 1) inhibit seed germination in *Lactuca sativa* [7]. In addition, nagilactone E exhibits antifungal activity against *Candida albicans*, *Pityrosporium ovale*, and the budding yeast *Saccharomyces cerevisiae* [8]. The activity of nagilactone E against *S. cerevisiae* depends on inhibition of 1,3-β glucan synthase [9]. As the activity is relatively weaker than that of other available antifungal drugs, such as micafungin and fluconazole [10], other drugs can be used in combination with nagilactone E to enhance its antifungal activity.

Anethole, a phenylpropanoid (Figure 1), is a major compound present in anise and fennel oils and exhibits antifungal activity against various fungi, including *S. cerevisiae* and *C. albicans* [8,11,12,13]. It alters the morphology of the filamentous fungus *Mucor mucedo* by inhibiting hyphal growth and the cell wall biosynthetic enzyme, chitin synthase [13]. In addition, it induces the production of reactive oxygen species and DNA fragmentation and induces apoptotic-like cell death in *S. cerevisiae* and *Aspergillus fumigatus* [14]. However, the antifungal activity of anethole, similar to nagilactone E, is relatively weaker than that of drugs available in the market [13].

Anethole enhances the antifungal activity of *n*-dodecanol, (*E*)-2-undecenal, polygodial, and nagilactone E in *S. cerevisiae* and *C. albicans* [8,15,16]. The antifungal effect of the model antifungal drug dodecanol against *S. cerevisiae* reduces with prolonged incubation because of the drug efflux activity of Pdr5p, a multidrug ATP-binding cassette (ABC) transporter [15,17,18]. Anethole suppresses *PDR5* transcription, thereby synergistically enhancing the antifungal activity of dodecanol [17,18].

In this study, we elucidated the genetic basis of the synergistic antifungal activity of nagilactone E and anethole against *S. cerevisiae* using deletion mutants. The effects of the combination of drugs on the yeast cell wall were determined using fluorescence and electron microscopy. Our results would benefit the practical application of nagilactone E as an antifungal agent.

## 2. Results

### 2.1. Antifungal Activity of Nagilactone E, Aenethole, and Their Combination

The minimum inhibitory concentration (MIC) assay was performed using the broth dilution method. The MIC values of nagilactone E, anethole, and their combination against *S. cerevisiae* and *C. albicans*, including fluconazole-resistant strains, are summarized in Table 1. The fractional inhibitory concentration (FIC) indices of drug combinations are also indicated. Because of the limited solubility of nagilactone E in *N*,*N*-dimethylformamide (DMF), a final concentration of more than 2000 μM was not tested. The MIC values of nagilactone E and anethole for *S. cerevisiae* after 72 h of incubation were 500 and 2500 μM, respectively. The FIC index of the drug combination was 0.3125, indicating a synergistic effect. The FIC indices for all *C. albicans* strains were less than 0.5625, indicating weak synergistic effects. These results suggest that fluconazole resistance does not affect the synergistic antifungal activity of the drug combination.

The cell viability of *S. cerevisiae* in the presence of nagilactone E, anethole, and their combination was determined using the number of colony-forming units (CFUs) (Figure 2). Anethole (312.5 μM) did not affect cell viability. Although nagilactone E (250 μM) reduced cell viability within 48 h of incubation, cell viability recovered by 72 h. In contrast, when anethole and nagilactone E were used in combination, the decrease in cell viability continued until 72 h. To summarize the above results, anethole extended the fungicidal activity of nagilactone E. Anethole showed synergy with the surfactant dodecanol (log of partition coefficient: logP = 4.31) via the inhibition of drug efflux [18]. Nagilactone E is also an amphiphilic substance (logP = 1.26, logP = 1.39 for *n*-pentanol). Thus, the drug may be continually transported out of the cell by multidrug ABC transporters located on the cell membrane. In addition, anethole possibly inhibits nagilactone E drug efflux, thereby sustaining its antifungal activity.

### 2.2. Identification of Genes Associated with Nagilactone-Efflux Using Gene-Deficient Strains

To identify the genes associated with multidrug resistance and susceptibility to nagilactone E, we evaluated the susceptibilities of single gene-deficient, double gene-deficient *pdr1*Δ *pdr3*Δ (BY25929), and 13 multi-gene-deficient (dTC063) strains and their parental BY4741 strains. After 72 h of incubation with nagilactone E, the cell turbidity of *pdr5*Δ, BY25929, and dTC063 cells was analyzed by absorbance measurements at 600 nm. The absorbance value was found to be less than 1.0, which was the adjusted value at the beginning of incubation (Figure 3). These results indicated that these three mutants are highly sensitive to nagilactone E. In addition, a decrease in cell turbidity indicated nagilactone-E-induced cell lysis. Cell lysis may be the result of a fragile cell wall due to the inhibition of β-glucan biosynthesis [9]. Moreover, nagilactone E was potentially effluxed by Pdr5p, as are amphiphilic and lipophilic xenobiotic compounds, including antibiotics [19]. Although double gene-deficient *pdr1*Δ *pdr3*Δ strains were susceptible to nagilactone E, single gene-deficient *pdr1*Δ or *pdr3*Δ strains were not. *PDR1* and *PDR3* are reported to be transcription factors required for *PDR5* transcription [20]. These results indirectly indicate that both *pdr1* and *pdr3* are required for the activation of *PDR5* transcription. Based on these results, we speculated that Pdr5p is an important multidrug efflux pump associated with the combinatorial antifungal activity of nagilactone E and anethole.

### 2.3. Effects of Anethole, Nagilactone E, and Their Combination on PDR5 Transcription

The transcript levels of *PDR5* may potentially influence nagilactone E efflux. To determine the effect of anethole on *PDR5* transcription in cells treated with nagilactone E, transcription levels of *PDR5* were quantified using reverse transcription quantitative polymerase chain reaction (RT-qPCR). Their transcription levels were normalized to 1.0 against transcript levels of the housekeeping gene *ACT1*. The levels were significantly higher in nagilactone-E-treated cells than those in untreated or anethole-treated cells. Interestingly, in the cells treated with the combination of drugs, the *PDR5* transcript level was more than 4-fold higher than that in the nagilactone-E-treated cells at 4 h after treatment. However, as incubation with the drugs continued, the level decreased gradually and finally reached almost the same level as that in untreated cells (Figure 4). These results indicated the possibility that when *PDR5* transcription is intensively induced by the drug combination, it is subsequently suppressed by anethole. In the drug-combination-treated cells, *PDR5* transcription was upregulated, especially at 4 and 24 h incubations, and was possibly translated to large amounts of Pdr5p. However, it is to be noted that transcription levels do not necessarily reflect the protein levels.

### 2.4. Visualization of Green Fluorescent Protein (GFP)-Tagged Pdr5p

To analyze Pdr5p protein levels in cells treated with nagilactone E, anethole, and their combination, we tagged Pdr5p with GFP at the C-terminus. The strain harboring GFP-tagged Pdr5p showed the same susceptibility toward the drugs as the parental BY4741 strain. No fluorescence was detected in the cytoplasm or vacuoles of the control cells treated with DMF. A slight fluorescence was detected in the anethole-treated cells (Figure 5). Strong fluorescence was detected on the membranes, especially plasma membranes, in nagilactone-E-treated cells only at 72 h (Figure 5). These results indicated that Pdr5p translation is induced in the presence of nagilactone E after 48 h of incubation. In contrast, almost no fluorescence was observed in cells treated with the drug combination throughout incubation. Therefore, anethole or nagilactone E probably restricted Pdr5p expression at the translational level.

In the time-kill assay (Figure 2), the viability of nagilactone-E-treated cells was recovered after 48 h of incubation, whereas a gradual decrease in the viability of drug-combination-treated cells was observed. This difference is probably because of an increase in Pdr5p levels after 48 h of incubation. In other words, an increase in Pdr5p levels caused by nagilactone E accumulation in the cells takes at least 48 h.

### 2.5. Visualization of β-Glucan and Mannan on the Cell Surface

Nagilactone E weakly inhibits β-glucan synthase and then causes a slight fragility in the fungal cell wall [9]. To determine whether anethole accelerates the inhibitory effect of nagilactone E, the cell wall components of treated and untreated cells were visualized by staining with aniline blue or fluorescently labeled ConA (FITC-ConA). Aniline blue and FITC-ConA are usually used to stain β-glucan and mannan present in the cell wall, respectively [21]. In the spheroplasts treated with zymolyase, no fluorescence was detected, indicating a lack of β-glucan (Appendix A). In contrast, fluorescence was observed around the cell circumference in all the cells treated with nagilactone E and/or anethole (Appendix A). These results indicated that nagilactone E and anethole target β-glucan and mannan in the cell wall. However, the fluorescence attributed to aniline blue was not uniform, and some portions of the cells were weakly or strongly stained after treatment with nagilactone E and a combination of both drugs. Strong fluorescence indicates the possibility of local β-glucan accumulation or perhaps extraordinary β-glucan biosynthesis. The localization of mannan in the nagilactone-E- and combination-treated cells was similar to that in the control cells. The inhibition of cell wall synthesis by such drugs possibly causes the decline in fluorescence. However, the effects of nagilactone E and the drug combination on the inhibition of β-glucan synthesis could not be confirmed based on β-glucan staining with fluorescent reagents.

### 2.6. Cell Surface Structure and Thickness of Cell Wall

If the architecture of the yeast cell wall was affected by nagilactone E upon combinatorial treatment with drugs, visible damage could be detected on the cell surface or thinning of the cell wall could be observed. To observe the changes in the cell wall structure, especially at the surface, we performed quick-freeze deep-etch electron microscopy.

Exponentially growing cells were incubated with or without drugs and prepared for microscopy. The shapes of the cells visualized by electron microscopy were similar to those of living cells visualized by optical microscopy (Figure 6A). The fibrous structures were observed to be intricately intertwined on the cell surface (Figure 6A, surface), and the thickness of the cell wall was approximately 100 nm (Figure 6A, fractured). In the fractured surface image, a two-layered structure was observed on the cell surface with entangled fibers on the outer layer. The inner side had high-density packed materials (Figure 6A fractured). The surface of the cells treated with nagilactone E or anethole alone was filamentous, similar to that of control cells. However, a few fiber-like structures were observed on the surface of cells treated with the drug combination (Figure 6B). Namely, the surface was largely smooth; moreover, the outermost layer seemed to be peeled off. These results indicated that the drug combination affects the integrity of the outermost layer of the cell wall. The cell wall structure was maintained until 72 h after treatment with nagilactone E alone.

The decrease in cell viability was the highest after 48 h of treatment with nagilactone E. Hence, we observed the cell surface after 48 h of treatment with nagilactone E or the drug combination. On the surface of cells treated with 500 μM (twofold of MIC), the density and number of fibers were apparently reduced (Figure 6C); however, we did not find any differences in the structures present on the cell surface and in fractured surface images of cells treated with 250 μM nagilactone E (MIC) or a combination of 250 μM nagilactone E and 312 μM anethole, similar to control cells.

We tested the effect of the β-glucan biosynthesis inhibitor, micafungin, on the cell surface structure. Prior to microscopic observation, the cells were incubated with 0.1~0.4 μM micafungin for 24 or 48 h. The MIC of micafungin was 0.2 μM after 48 h incubation. As the concentration of micafungin increased, the filamentous structures on the cell surface gradually decreased and became rough at 0.4 μM (Figure 7A). The density of fibers present on the cell surface after treatment with 0.4 μM for 48 h was similar to that of fibers treated with 500 μM nagilactone E. The tangled fibers with gaps were observed on the cell surface and were different from the cell surface of drug-combination-treated cells (Figure 6B). The cell walls of micafungin-treated cells were stained with aniline blue and FITC-ConA (Appendix A). This supported the results obtained by electron microscopy.

The cell wall of *S. cerevisiae* consists of three components: chitin, β-glucan, and mannoproteins. The content of chitin is lower than that of other components, except in the chitin ring, which is associated with septum formation during budding [22]. Among all the cell wall components, β-glucan and mannoproteins could not be visually distinguished on the basis of the cell surface characteristics observed in the electron micrographs. Therefore, we attempted to distinguish them by treating the cells with mannanase and/or proteinase K. After treatment, we stained the cells with FITC-ConA that targets mannan and mannoproteins. The cell walls of enzyme-treated cells were uniformly stained. Thus, we found no difference in the mannan content of the cells (Appendix A). Differences in the thickness of fibers were observed in cells treated with or without proteinase K, whereas no significant differences were observed in mannanase-treated cells (Figure 8). The fibers present on the cell surface appeared white in the images. The ratio of the black to the white area was calculated after the conversion of monochrome binary images (Figure 9). In the images of the proteinase-K- and drug-combination-treated cells, the percentage of the black area was high. This observation indicated that the fibers, which appear as white areas, were reduced or became thin upon treatment. However, we could not distinguish between β-glucan and mannoproteins in the electron micrographs.

Finally, we observed the surface of spheroplast cells prepared by zymolyase treatment. Zymolyase contains protease, β-1,3-glucanase, and mannanase. The surface of zymolyase-treated cells was smooth, and few filamentous structures were observed (Figure 7B). Similar results were also obtained for drug-combination-treated cells (Figure 6B), wherein the cells contained reduced β-glucan and mannan content. Treatment with zymolyase might have caused a collapse of the cell wall, thereby entirely removing it. Therefore, the smooth surface observed in zymolyase-treated cells reflects the presence of residual cell wall components in the plasma membrane environment. In contrast, in the case of the drug-combination-treated cells, the smooth surface possibly reflects a lack of intertwined fibers in the outermost layer of the cell wall.

## 3. Discussion

Nagilactone E exhibits antifungal activities against nonpathogenic *S. cerevisiae* and the human pathogen *C. albicans* by inhibiting 1,3-β glucan synthase and thereby affecting the fragility of the cell wall [9,21]. However, its activity is weaker than that of antifungal drugs present in the market. Therefore, the antifungal effect of nagilactone E in combination with other drugs needs to be checked, preferably with natural antifungal agents. Anethole is known to assist other drugs such as *n*-dodecanol and shows synergistic antifungal activity by inhibiting *PDR5* transcription and affecting drug efflux pump activity [17]. In this study, we demonstrated the synergistic effects of nagilactone E and anethole in combination and elucidated the underlying mechanism of their synergistic action.

The FIC indices (Table 1) and results of the time-kill assay (Figure 2) confirmed the synergistic effects of nagilactone E and anethole in combination. In addition, the strain lacking the *PDR5* gene was more susceptible to nagilactone E than its parental strain (Figure 3). Nagilactone E increased *PDR5* transcription and anethole suppressed *PDR5* upregulation in the drug-combination-treated parental strain after 72 h (Figure 4). Fluorescence derived from GFP was intensely detected in nagilactone-E-treated cells but not in combination-treated cells (Figure 5). To summarize these results, nagilactone E induced Pdr5p overexpression. This could explain why the growth of nagilactone-E-treated cells recovered as incubation proceeded. In other words, anethole inhibits *PDR5* transcription and maintains nagilactone E within cells. However, *PDR5* transcription in combination-treated cells considerably increased at 4 h (Figure 4), and the reason behind this is currently unknown. The inhibition of the transcription may be potentiated as the increasing intracellular concentration of anethole. In addition, several transcription products were probably not translated; therefore, no fluorescence from GFP was observed in combination-treated cells. In addition, in nagilactone-E-treated cells, a time lag was observed between the increase in *PDR5* transcription and increase in fluorescence intensity (Figure 4 and Figure 5). Although constant levels of transcription were present, as measured by RT-qPCR, *PDR5* expression was checked after 72 h of incubation. Although the mechanisms of post-transcriptional regulation, such as increased translational efficiency of eukaryotes, have been reported previously [23,24,25,26,27], post-transcriptional regulation via mRNA degradation and other regulatory mechanisms of *PDR5* are unclear. In the control cells, weak fluorescence was observed, even in the absence of transcription, and can be because of translation of all the residual transcription products. Therefore, protein levels and localization of Pdr5p are to be interpreted, and not the transcription level.

To visualize β-glucan in the cell wall of combination-treated cells, we stained the cells with aniline blue. No fluorescence was detected in zymolyase-treated spheroplasts, indicating the absence of β-glucan (Appendix A). Fluorescence along the perimeter of cells was detected without interruption in combination-treated and nagilactone-E-treated cells (Appendix A). The other major component, mannan, was examined in a similar way using FITC-ConA (Appendix A). The results indicated the presence of β-glucan and mannan tightly surrounding the cell walls. From these results, we could not confirm the apparent inhibition of cell wall β-glucan synthesis. As fluorescence intensity was related to the amount of the target mannan and β-glucan, the intensity did not indicate the strength of the cell wall.

Next, we analyzed the effects of nagilactone E on the cell wall of drug-treated yeast cells using quick-freeze deep-etch electron microscopy. The cell wall consists of mannan and glucan, the primary components of the cell wall [21,28,29,30,31]. Under normal growth conditions, micrographs of the fractured surfaces revealed planes behind the cell wall, which were exposed by deep etching, and showed regions of aggregations of granular objects, possibly mannoproteins [32] (Figure 6A). The planes were considered to be the surface of the plasma membrane. In drug-combination-treated cells (Figure 6B), no fibers were observed on the cell surface, indicating an abnormal cell wall architecture, similar to the surface of the zymolyase-treated cells. The combination-treated cells showed aniline blue staining, whereas zymolyase-treated cells did not. The visible stained structures on the surface after treatment with a combination of drugs did not include the plasma membrane. Our results obtained using cells stained with FITC-ConA and aniline blue indicated that mannanase and/or proteinase K treatment cannot completely remove mannoproteins, unlike zymolyase treatment. However, no significant difference was observed in the electron micrographs. In contrast, similar fibrous structures of the outermost layer were observed in the vegetative cells of the fission yeast, *Schizosaccharomyces pombe* [33]. These fibers are known to consist of galactomannan in *S. pombe* [34]. This indicates that the fibers in the outermost layers of the *S. cerevisiae* cell wall are possibly mannan, as determined in previous reports [35,36]. Although small changes were detected on the cell surface by enzyme treatments including proteinase K (Figure 8 and Figure 9), these changes could not lead us to conclude that the outermost layer of the cell wall only consisted of mannoproteins. Further investigations are needed to reveal the detailed structure of the cell wall of budding yeast, including the top surface fibers.

Contradictory results were obtained in aniline blue staining and electron microscopy. Although cell surface structures with a few fibers were observed in combination-treated and zymolyase-treated cells, the combination-treated cells showed aniline-blue-positive staining, whereas zymolyase-treated cells remained unstained. This could be explained by morphological differences observed in the cells based on the phase-contrast images (Appendix A). Elliptical yeast forms were observed in combination-treated cells, and circular forms were observed in zymolyase-treated cells. The circular form probably indicates a complete loss of the cell wall. The ellipse form indicates the presence of residual cell wall components. This difference in morphology and the results of electron microscopy collectively indicated that the cells treated with the drug combination were not spheroplasts; however, the cell wall underwent significant alterations upon treatment.

Although nagilactones are known to be plant growth inhibitors, nagilactone E only inhibits fungi. Various phenylpropanoids, including anethole, probably enhance the antifungal activity of nagilactone E against *S. cerevisiae*, *C. albicans*, and *P. ovale*. In combination with anethole, nagilactone E exhibits a synergistic effect. As a result, the cell walls became brittle. The drug combination induced cell lysis under low osmotic conditions such as malt extract (ME) medium, and then anethole potentiated the activity of nagilactone E. Synergistic effects are expected to be observed in cells treated with nagilactone E in combination with other phenylpropanoids, probably inducing the inhibition of drug efflux, similar to anethole.

Nagilactone E weakly inhibits β-glucan synthase in fungi [9] and is transported by the drug efflux pump. The antifungal activity of the drug is relatively weak; however, it can be used as an antifungal agent because of its selective action on fungi, including human pathogens *C. albicans* and *Aspergillus* spp., in addition to *S. cerevisiae* [37,38]. Furthermore, enhancing the weak antifungal activity of nagilactone E may help us to use it to restrict fungal food spoilage.

## 4. Materials and Methods

### 4.1. Strains and Culture Conditions

The parental strain *S. cerevisiae* BY4741 (*MATa*, *ura3*Δ*0*, *leu2*Δ*0*, *met15*Δ*0*, and *his3*Δ*1*) and its derived knockout strains (*pdr1*Δ/YGL013C, *pdr3*Δ/YBL005W, *pdr5*Δ/YOR153W, *pdr8*Δ/YLR266C, *pdr10*Δ/YOR328W, *pdr11*Δ/YIL013C, *pdr12*Δ/YPL058C, *pdr15*Δ/YDR406W, *adp1*Δ/YCR011C, *aus1*Δ/YOR011W, *snq2*Δ/YDR011W, *yrs1*Δ/YGR281W, and *yrr1*Δ/YOR162C) were purchased from the Yeast Knockout Strain Collection (Thermo Scientific Open Biosystems, Waltham, MA, USA). *S. cerevisiae* dTC063 (*pdr1*Δ, *pdr3*Δ, *pdr5*Δ, *pdr8*Δ, *pdr10*Δ, *pdr11*Δ, *pdr12*Δ, *pdr15*Δ, *adp1*Δ, *aus1*Δ, *snq2*Δ, *yrs1*Δ, and *yrr1*Δ) was gifted by Prof. Usui Takeo, University of Tsukuba (Tsukuba, Japan). *S. cerevisiae* BY25929 (*pdr1*Δ *pdr3*Δ) was provided by the National BioResource Project -Yeast (Osaka, Japan). The parental strain *C. albicans* NBRC1061 (wild type) was obtained from the Biological Resource Center, National Institute of Technology and Evaluation (Tokyo, Japan). *C. albicans* IFM46910 and IFM54354 (fluconazole resistant) were provided by the Medical Mycology Research Center, Chiba University (Chiba, Japan). The yeast cells were grown in 2.5% ME (*m*/*v*; Oriental Yeast, Tokyo, Japan) broth at 30 °C, unless otherwise stated.

### 4.2. Chemicals

Nagilactone E was taken from the laboratory stock [8]. *trans*-Anethole (anethole) was purchased from Sigma-Aldrich (St. Louis, MO, USA). Micafungin was gifted by Astellas Pharma Inc. (Tokyo, Japan). Zymolyase 20T was purchased from Nacalai Tesque (Kyoto, Japan). Mannanase was gifted by Amano Enzyme Inc. (Nagoya, Japan). FITC-ConA was purchased from Seikagaku Corporation (Tokyo, Japan). Aniline blue dye and DMF were purchased from Wako Pure Chemicals (Osaka, Japan). Proteinase K was purchased from Merck (Darmstadt, Germany). The drugs were dissolved in their respective solvents before starting the experiments. Nagilactone E, *trans*-anethole, and micafungin were dissolved in DMF. Aniline blue dye was dissolved in 2 mM 4-(2-hydroxyethyl)-1-piperazineethanesulfonic acid (HEPES, pH 7.6) containing 150 mM NaCl and 1.2 M sorbitol. FITC-ConA was dissolved in phosphate-buffered saline (PBS) consisting of 1.4 M NaCl, 200 mM Na_2_HPO·12H_2_O, 27 mM KCl, and 15 mM KH_2_PO_4_.

### 4.3. Antifungal Susceptibility Assay

The antifungal susceptibility assay was performed, as previously described [39]. Serial twofold dilutions of antifungal drugs were prepared in DMF, and 30 μL of the drug solution was added to 3 mL of yeast peptone dextrose (YPD) medium consisting of 1% Bacto-yeast extract (BD, Franklin Lakes, NJ, USA), 2% Bacto-peptone (BD), and 2% D-glucose in a test tube (diameter of 10 mm). The exponentially growing yeast cells of *S. cerevisiae* BY4741 and *C. albicans* NBRC1061, IFM46910, and IFM54354, were inoculated in the medium at a final cell density of 1.0 × 10^6^ CFU mL^−1^. The cultures were incubated without shaking at 30 °C for 72 h. After incubation, the MIC was determined, which was defined as the lowest concentration of the test compound at which no visible yeast growth was detected.

### 4.4. Time-Kill Assay

Exponentially growing cells (1.0 × 10^6^ CFU mL^−1^) of *S. cerevisiae* BY4741 were incubated without shaking at 30 °C for 72 h in ME broth containing anethole, nagilactone E, or their combination. Aliquots of cell suspensions were withdrawn, diluted, and spread on YPD agar medium. The viable cell count was determined based on the number of CFUs after incubation for 48 h at 30 °C.

### 4.5. Measurement of Cell Turbidity in Gene Deletion Strains Related to Drug Efflux

Exponentially growing yeast cells of each strain (1.0 × 10^7^ CFU mL^−1^) were incubated with shaking at 30 °C for 72 h in ME broth containing 250 μM nagilactone E. After incubation, the absorbance, as a measure of cell turbidity, was measured using a UV-2450 spectrophotometer (Shimadzu, Kyoto, Japan) at wavelength of 600 nm.

### 4.6. RNA Extraction and RT-qPCR

Total RNA was extracted from *S. cerevisiae* BY4741 cells using the RNeasy Mini Kit (Qiagen, Hilden, Germany) according to the manufacturer’s instructions. Briefly, yeast cells treated with anethole, nagilactone E, or their combination were harvested by centrifugation at 3700× *g* for 10 min and then lysed using 6.2 mg mL^−1^ of zymolyase 20T. The RNA samples were purified using the columns provided in the kit and were then treated with DNase. The RNA samples were reverse-transcribed to generate complementary DNA using ReverTra Ace (Toyobo, Osaka, Japan). The CDS primer (5′-AAGCAGTGGTAACAACGCAGAGATACTTTTTTTTTTTTTTTTTTTTTTTTTTTTTTVN-3′) was used for the reverse transcription reaction. RT-qPCR was performed using SsoAdvanced Universal SYBR Green Supermix (Bio Rad, Hercules, CA, USA) with complementary DNA as a template using the CFX Connect Real-Time PCR Detection System (Bio Rad, Hercules, CA, USA). The qPCR reaction mixture (20 μL) contained 10 μL of 2× Supermix, 5 ng of the template, and 5 pM of each of forward and reverse primers. The cycling profile was 3 min at 95 °C, followed by 40 cycles of 10 s at 95 °C, and 30 s at 55 °C. The relative expression of *PDR5* was normalized to that of the housekeeping gene, *ACT1*, as an internal positive control. The primers used in this study are listed in Table 2.

### 4.7. Visualization of GFP-Pdr5p Fusion Protein

Yeast cells (1.0 × 10^7^ CFU mL^−1^) were incubated in ME broth with anethole, nagilactone E or their combination for 72 h. Aliquots were collected at the indicated times and centrifuged at 6800× *g* at 25 °C for 3 min. The harvested cells were washed twice with PBS and observed under a BX51 microscope (Olympus, Tokyo, Japan). The excitation and emission wavelengths for fluorescence visualization were 488 and 507 nm, respectively.

### 4.8. Aniline Blue Staining

Exponentially growing yeast cells (1.0 × 10^7^ CFU mL^−1^) of *S. cerevisiae* BY4741 were incubated without shaking at 30 °C for 72 h in ME broth with 1.2 M sorbitol containing 250 μM nagilactone E and/or 312.5 μM anethole. After incubation, the cells were harvested by centrifugation, washed twice with 2 mM HEPES buffer (pH 7.6) containing 150 mM NaCl and 1.2 M sorbitol, and resuspended in a small amount of HEPES buffer, and then 0.5% aniline blue dye (1/10 amount) was added and the cells were allowed to stand for 5 min. The cells were observed under a BX51 microscope (Olympus). The excitation and emission wavelengths for fluorescence visualization were 370 and 430 nm, respectively.

### 4.9. Creating Spheroplasts

Exponentially growing yeast cells (1.0 × 10^7^ CFU mL^−1^) of *S. cerevisiae* BY4741 were incubated without shaking at 30 °C for up to 4 h in ME broth containing 1.2 M sorbitol. The cells were washed twice with HEPES containing 1.2 M sorbitol and then lysed using 15.6 mg mL^−1^ of zymolyase 20T at 30 °C for 30 min with shaking in HEPES. Next, 0.5% aniline blue was added, and the cells were observed under the microscope.

### 4.10. Visualization of the Cell Wall Using FITC-ConA

Exponentially growing yeast cells (1.0 × 10^7^ CFU mL^−1^) were incubated without shaking at 30 °C for 4 h in ME broth with 1.2 M sorbitol containing 250 μM nagilactone E and/or 312.5 μM anethole and then washed twice with PBS containing 1.2 M sorbitol. The cells were washed with PBS, stained with FITC-ConA for 30 min, fixed with 4% formalin for 30 min, and then observed under a BX51 microscope (Olympus). The excitation and emission wavelengths for fluorescence visualization were 495 and 514 nm, respectively. The cells treated with 70 mg mL^−1^ of mannanase and/or 1 μL mL^−1^ of proteinase K were stained with FITC-ConA at 37 °C for 30 min with shaking in PBS containing 1.2 M sorbitol.

### 4.11. Quick-Freeze Deep-Etch Replica Electron Microscopy

Exponentially growing yeast cells (1.0 × 10^6^ CFU mL^−1^) of *S. cerevisiae* BY4741 were incubated in ME broth containing testing compounds for the indicated time without shaking. The cells were washed twice with HEPES buffer and harvested by centrifugation. The cells were mixed with a slurry that included mica flakes, placed on a rabbit lung slab, and frozen using CryoPress (Valiant Instruments, St. Louis, MO, USA) that was cooled using liquid helium [33,40]. The slurry was used to retain the appropriate volume of water before freezing. The specimens were fractured and etched for 15 min at −104 °C using a JFDV freeze-etching device (JEOL Ltd., Akishima, Japan). The exposed cells were rotary-shadowed using platinum at an angle of 20° and a thickness of 2 nm and backed with carbon. The replicas were floated off on full-strength hydrofluoric acid, rinsed with water, cleaned with commercial bleach, rinsed with water, and kept on copper grids, as described. The replica specimens were observed using a JEM-1010 transmission electron microscope (TEM, JEOL, Tokyo, Japan) at 80 kV, equipped with a FastScan-F214 (T) charge-coupled device (CCD) camera (TVIPS, Gauting, Germany). The cells (1.0 × 10^7^ CFU mL^−1^) were also treated with zymolyase 20T, mannanase, and/or proteinase K in 2 mM HEPES buffer (pH 7.6) containing 150 mM NaCl and 1.2 M sorbitol at 30 °C (zymolyase 20T) or 37 °C (mannanase or proteinase K) for 30 min with shaking. After treatment with mannanase and/or proteinase K, the cells were fixed with 4% formalin, washed twice with water, centrifuged, and analyzed, as described above.

### 4.12. Image Analyses for Cell Surface Damage

All the images obtained using TEM were inverted to black-and-white binary using ImageJ 1.51j8 software. All the images were processed with a filter of smooth command six times and then converted to monochrome binary using a threshold command at the maximum threshold value set to 120 in 256 gradations. The number and area of black particles with more than 100 pixels in the images were then analyzed. For each image, the number and area ratios of the black particles were calculated.

## Figures and Tables

**Figure 1 antibiotics-10-00537-f001:**
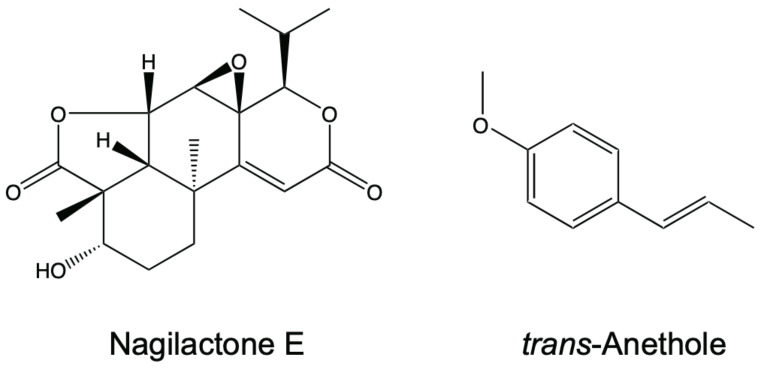
Chemical structures of nagilactone E and *trans*-anethole.

**Figure 2 antibiotics-10-00537-f002:**
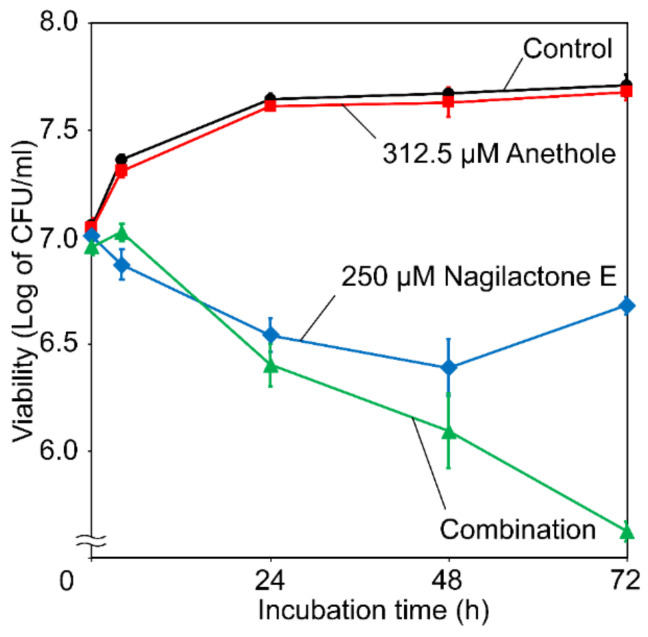
Cell viability in the presence of anethole, nagilactone E, or their combination. Exponentially growing cells of *S. cerevisiae* BY 4741 at 30 °C in YPD medium supplemented with 312.5 μM anethole (■), 250 μM nagilactone E (◆), or 312.5 μM anethole + 250 μM nagilactone E (▲). Closed circles (●) denote controls (treatment without drugs). Data are expressed as the mean ± standard deviation (n = 3).

**Figure 3 antibiotics-10-00537-f003:**
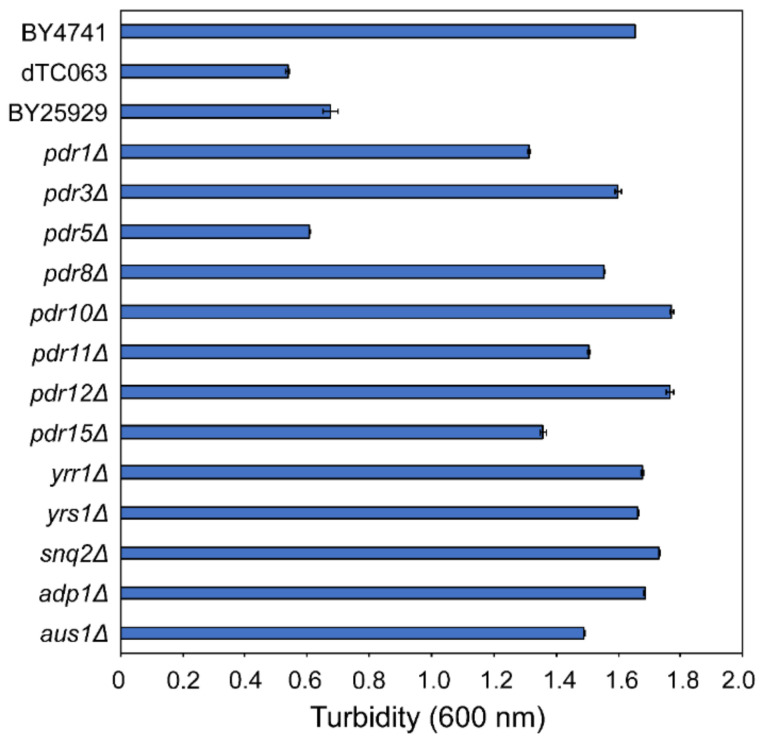
Susceptibility to nagilactone E in *S. cerevisiae* mutants lacking genes associated with multidrug resistance. BY4741, dTC063, and BY25929 are parental, 13 multi-gene-deficient, and double gene-deficient (*pdr1*Δ *pdr3*Δ) strains, respectively. Exponentially growing cells were grown at 30 °C with 72 h shaking in 2.5% ME with 250 μM nagilactone E. Data are expressed as the mean ± standard deviation (n = 3).

**Figure 4 antibiotics-10-00537-f004:**
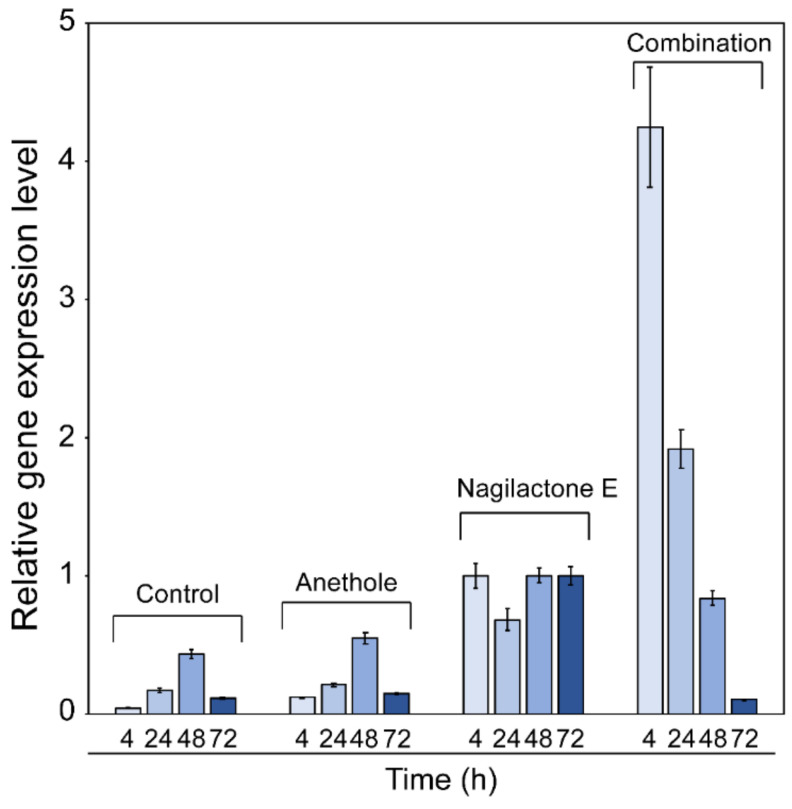
Relative gene expression of *PDR5* in *S. cerevisiae* BY4741. Exponentially growing cells were grown at 30 °C in 2.5% ME treated with 312.5 μM anethole, 250 μM nagilactone E, or their combination. Control denotes treatment without drugs. Total RNA was then extracted and analyzed by RT-qPCR. Gene expression is shown relative to *ACT1* expression. Data are expressed as the mean ± standard deviation (n = 3).

**Figure 5 antibiotics-10-00537-f005:**
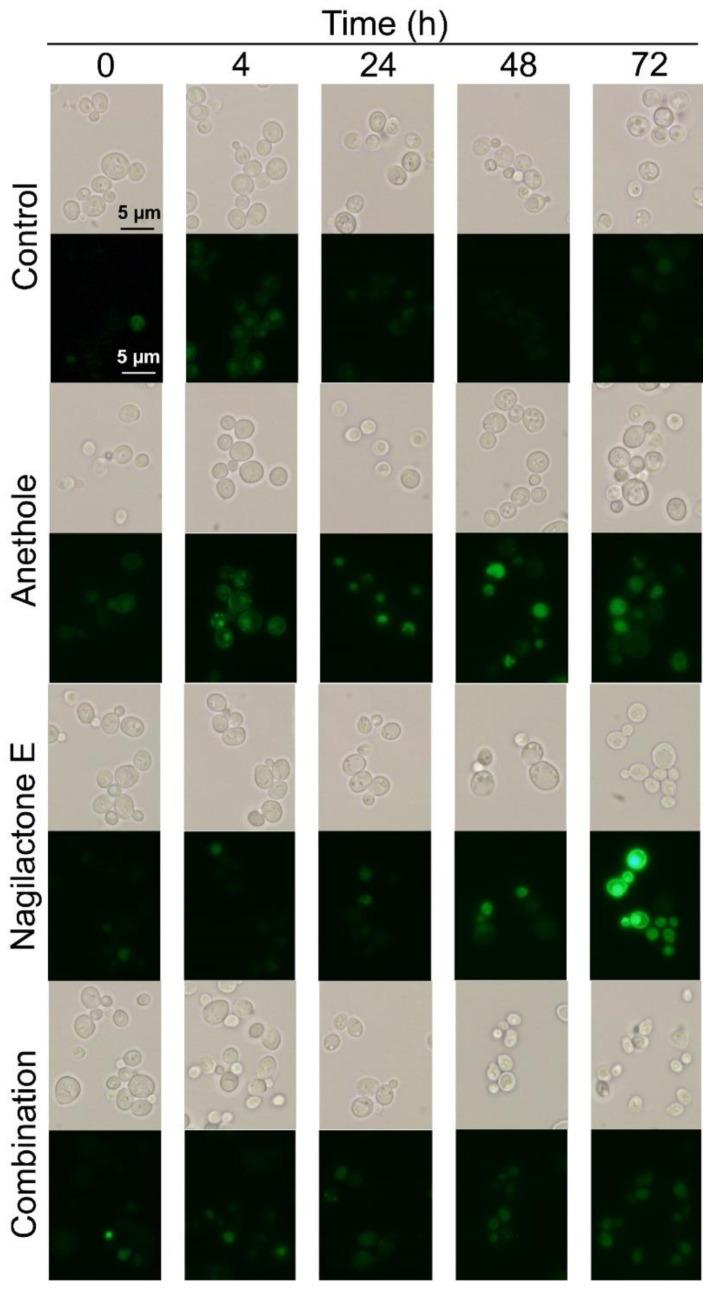
Phase-contrast and fluorescent micrographs of cells expressing Pdr5p tagged with green fluorescent protein. Exponentially growing cells were grown at 30 °C in 2.5% ME treated with 312.5 μM anethole, 250 μM nagilactone E, or their combination. Control denotes treatment without drugs. Refer to Appendix A for the original uncropped photos of cells after 72 h of incubation with and without drugs.

**Figure 6 antibiotics-10-00537-f006:**
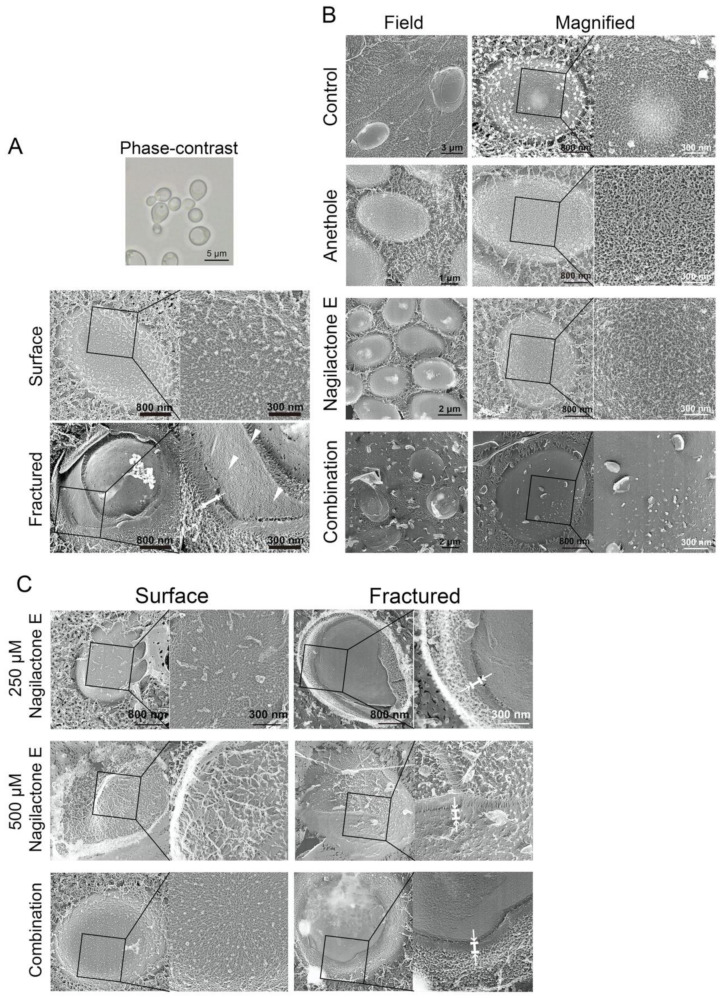
Quick-freeze deep-etch replica images of *S. cerevisiae* BY4741 cells. (**A**) The surface and fractured images of yeast cells (magnified). Exponentially growing cells without drug treatment were observed. (**B**) The surface of yeast cells in field and magnified images. Exponentially growing cells were grown at 30 °C for 72 h in 2.5% ME treated with 312.5 μM anethole, 250 μM nagilactone E, or their combination. Control denotes treatment without drugs. (**C**) The surface and fractured images of yeast cells (magnified). Exponentially growing cells were grown at 30 °C for 48 h in 2.5% ME treated with 250 and 500 μM nagilactone E and 250 μM nagilactone E combined with 312.5 μM anethole (combination).

**Figure 7 antibiotics-10-00537-f007:**
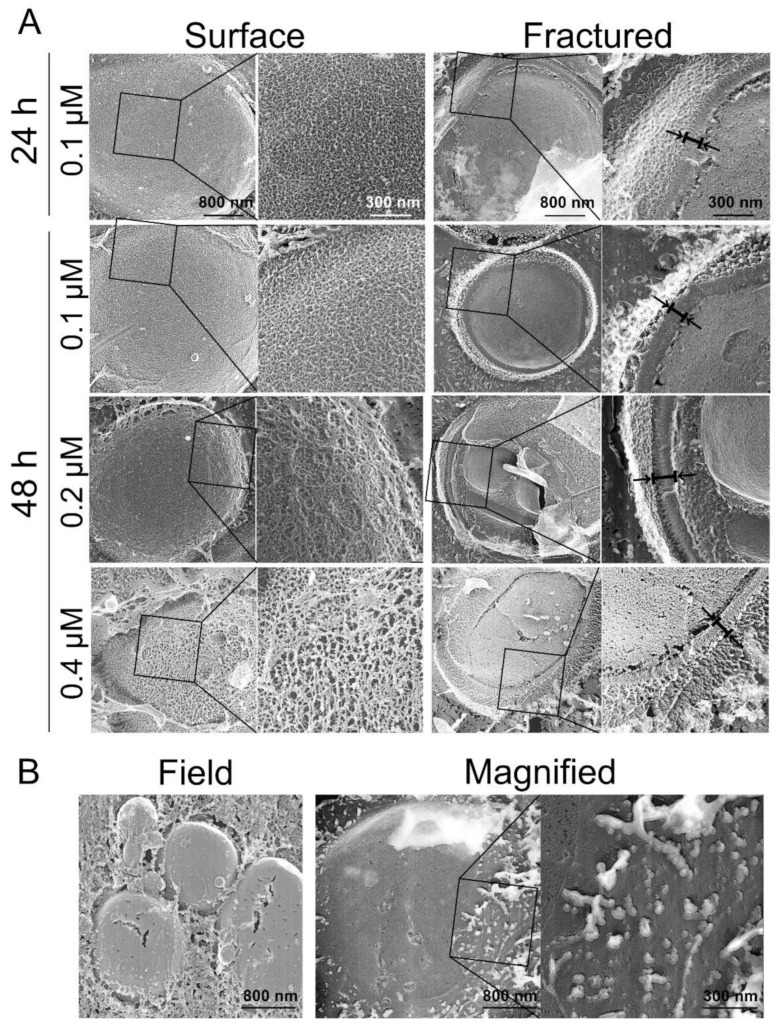
The surface and fractured images of micafungin- or zymolyase-treated yeast cells. (**A**) Exponentially growing cells were cultured in 2.5% ME treated with micafungin at 30 °C. (**B**) Exponentially growing cells were treated with zymolyase 20T at 30 °C for 30 min.

**Figure 8 antibiotics-10-00537-f008:**
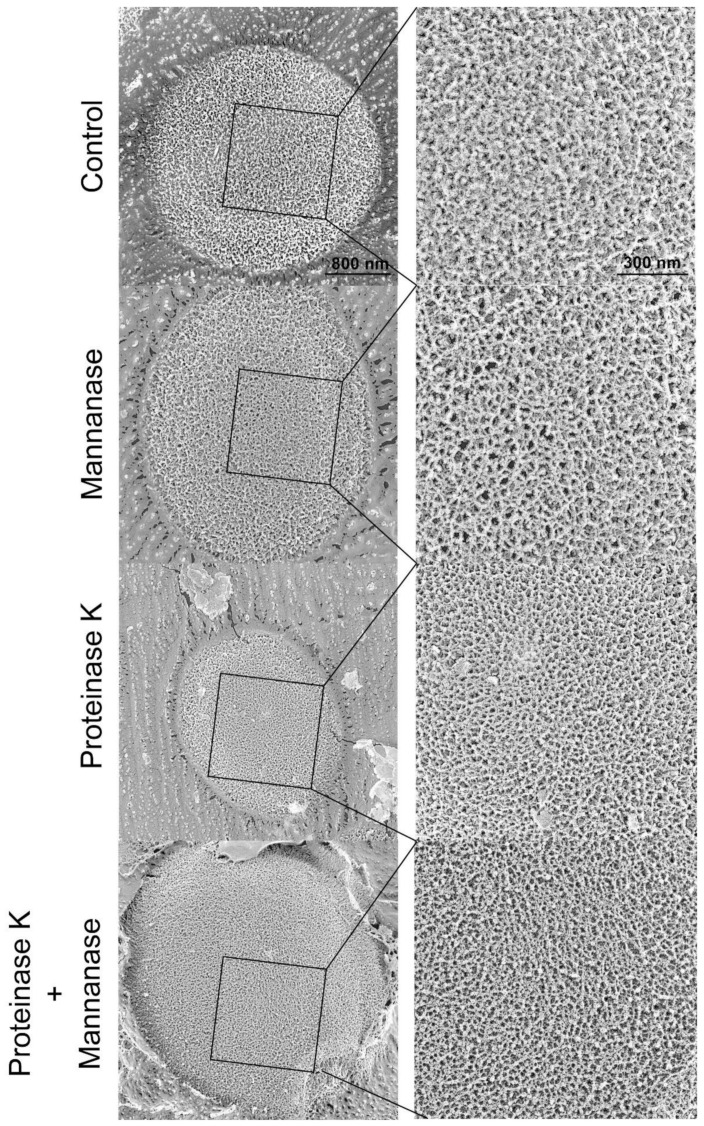
The surface of yeast cells treated with mannanase and/or proteinase K in field and magnified images. Exponentially growing cells were treated with mannanase and/or proteinase K at 37 °C for 30 min.

**Figure 9 antibiotics-10-00537-f009:**
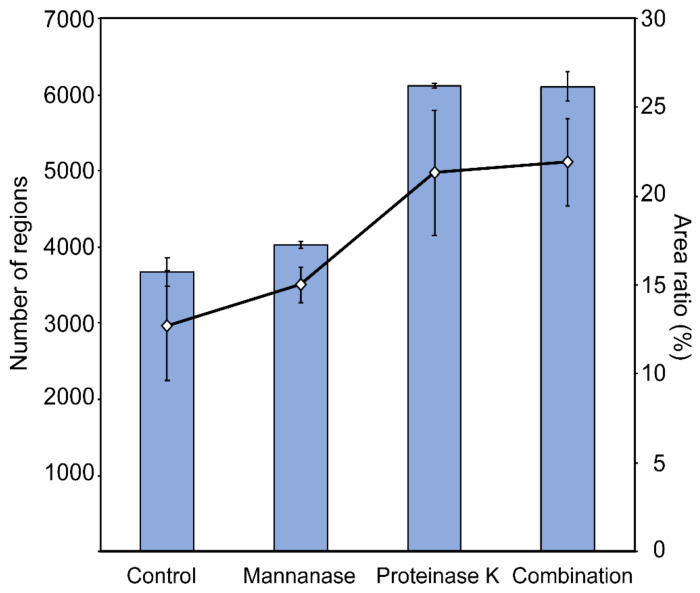
Quantification of the total dent area on the cell surface. The surface images in Figure 8 were converted to black-and-white binary. The number of black areas with an area of 100 pixels or more (bar chart) was counted, and the percentage of the area it occupies (line chart) was also calculated. The increase in the black area indicates an increase in the dent area on the cell surface. Data are expressed as the mean ± standard deviation (n = 3).

**Table 1 antibiotics-10-00537-t001:** MICs of nagilactone E, anethole, or their combination against *S. cerevisiae* or *C. albicans.*

Strain		MIC ^a^ (μM) at 72 h		FIC ^b^ Index
		Nagilactone E	Anethole	
*S. cerevisiae* BY4741	Alone	500	-	-
		-	2500	-
	Combination	31.3	625	0.31
*C. albicans* NBRC1061	Alone	1000	-	-
		-	1250	-
	Combination	62.5	625	0.56
*C. albicans* IFM46910 ^c^	Alone	>1000	-	-
		-	1250	-
	Combination	250	313	<0.5
*C. albicans* IFM54354 ^c^	Alone	>1000	-	-
		-	1250	-
	Combination	62.5	625	<0.56

^a^ Minimum inhibitory concentration. ^b^ Fractional inhibitory concentration. The FIC index for the drug combination was calculated as (MICa combination/MICa alone) + (MICb combination/MICb alone), where a and b are the two compounds tested. The FIC index values are significant values obtained from the checkerboard matrix. FIC indices were used to define the interaction of combined compounds: synergistic (X < 0.5), additive (0.5 < X < 1), indifferent (1 < X < 4), or antagonistic (X > 4). ^c^ Fluconazole-resistant strains.

**Table 2 antibiotics-10-00537-t002:** Primers used for RT-qPCR.

Primer Name	Sequence (5′-3′)
*ACT1*-F	ATGGTCGGTATGGGTCAAAA
*ACT1*-R	AACCAGCGTAAATTGGAACG
*PDR5*-F	GTTGCCTAAACCCAGGTGAA
*PDR5*-R	GTTGCCTAAACCCAGGTGAA

## Data Availability

The data presented in this study are available on request from the corresponding author.

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
