# Peer review of "Involvement of a Multidrug Efflux Pump and Alterations in Cell Surface Structure in the Synergistic Antifungal Activity of Nagilactone E and Anethole against Budding Yeast Saccharomyces cerevisiae"

_antibiotics, 2021, doi:10.3390/antibiotics10050537_

Round 1

Reviewer 1 Report

In the present paper Ueda et al investigate the antifungal effect of plant derived nagilactone E, which inhibits fungal glucan synthesis when used in combination with anethole. While the positive effect of the drug combination was described before, the mechanism remained unclear. Here, Ueda et al. show that the Pdr5 drug efflux pump is important for nagilactone E resistance in Saccharomyces cerevisiae and that the combination with anethole impairs expression of a Pdr5-GFP fusion protein.  In addition, cell wall structure changes are reported, which indicate enhanced cell wall damage. The paper is well prepared and presents interesting results that improve the understanding of the positive antifungal effect of the combined nagilactone E/anethole application. Some points should be considered for further improvement and clarification.

Line 19/20: This appears a bit oversimplified. Transcription of PDR5 was not generally inhibited by the drug combination. In fact, it was strongly induced relative to individual application of nagilactone E or anethole at earlier time points (Figure 4). Please clarify.

2.4. (line 157f): In this part, Pdr5-GFP fusion protein is detected by fluorescence microscopy and conclusions about translation efficiency drawn by considering the separate mRNA expression analysis shown in Figure 4. I would recommend to make less definitive statements here since the microscopy data are only qualitative, showing representative images with only few cells. A more direct way to analyze this would be to detect mRNA and Pdr5-GFP fusion protein levels from identical cultures (harvest cells, split and isolate mRNA/total protein which are then used for qPCR and Western blotting). This is also relevant for the discussion, line 317f. Here it should again be taken into consideration that fluorescence microscopy data are only qualitative and that the information about mRNA content and Pdr5-GFP fusion protein content is not derived from identical samples. In fact, even the strains used are different (wild type BY4741 for mRNA data, Pdr5-GFP tagged strain for fluorescence microscopy).

2.5. Line 193f: “However, the effects of nagilactone E and the drug combination on the inhibition of β-glucan synthesis could not be confirmed based on β-glucan staining with fluorescent reagents.” If this is the conclusion from this section, why does it need to be presented? Consider removing or inclusion as supplementary information. Currently, Fig.6 spans two pages, which should be avoided.

Line 48/49: It could be misunderstood that fluconazole and micafungin also inhibit glucan synthase.

Figure 1: Consider showing the chemical structures in landscape mode (next to each other)

Author Response

In the present paper Ueda et al investigate the antifungal effect of plant derived nagilactone E, which inhibits fungal glucan synthesis when used in combination with anethole. While the positive effect of the drug combination was described before, the mechanism remained unclear. Here, Ueda et al. show that the Pdr5 drug efflux pump is important for nagilactone E resistance in Saccharomyces cerevisiae and that the combination with anethole impairs expression of a Pdr5-GFP fusion protein.  In addition, cell wall structure changes are reported, which indicate enhanced cell wall damage. The paper is well prepared and presents interesting results that improve the understanding of the positive antifungal effect of the combined nagilactone E/anethole application. Some points should be considered for further improvement and clarification.

The English of our manuscript has been proofread by Editage (https://www.editage.jp/).

Line 19/20: This appears a bit oversimplified. Transcription of PDR5 was not generally inhibited by the drug combination. In fact, it was strongly induced relative to individual application of nagilactone E or anethole at earlier time points (Figure 4). Please clarify.

Line 19-20: Thank you for your indication. It was rewritten as the description that it gradually decreased during prolonged incubation.

2.4. (line 157f): In this part, Pdr5-GFP fusion protein is detected by fluorescence microscopy and conclusions about translation efficiency drawn by considering the separate mRNA expression analysis shown in Figure 4. I would recommend to make less definitive statements here since the microscopy data are only qualitative, showing representative images with only few cells. A more direct way to analyze this would be to detect mRNA and Pdr5-GFP fusion protein levels from identical cultures (harvest cells, split and isolate mRNA/total protein which are then used for qPCR and Western blotting). This is also relevant for the discussion, line 317f. Here it should again be taken into consideration that fluorescence microscopy data are only qualitative and that the information about mRNA content and Pdr5-GFP fusion protein content is not derived from identical samples. In fact, even the strains used are different (wild type BY4741 for mRNA data, Pdr5-GFP tagged strain for fluorescence microscopy).

Line 159-161: We described that the susceptibility of the parent strain and the GFP strain was the same. 

Supplementary Fig. 1: By adding the original photographs of Figure 5 to the supplementary figure, we have eliminated the possibility of being a typical example with a small number of cells.

2.5. Line 193f: “However, the effects of nagilactone E and the drug combination on the inhibition of β-glucan synthesis could not be confirmed based on β-glucan staining with fluorescent reagents.” If this is the conclusion from this section, why does it need to be presented? Consider removing or inclusion as supplementary information. Currently, Fig.6 spans two pages, which should be avoided.

As the supplementary figure 2, Figure 6 was moved out of the manuscript. However, the results are left in the manuscript as they are necessary for discussion on the structure of cell surface.

Line 48/49: It could be misunderstood that fluconazole and micafungin also inhibit glucan synthase.

Line 48-49: The description was corrected so that it is not misunderstood.

Figure 1: Consider showing the chemical structures in landscape mode (next to each other)

Thank you for your indication. I fixed it according to the comments.

Reviewer 2 Report

Lines  84-85 It is unclear how the data on the sensitivity of yeast to the test substances lead to the conclusion: «These results suggest that human pathogenicity and flucon- 84 azole resistance does not affect the synergistic antifungal activity of the drug combination.»

Fig.4. - It would be interesting to discuss in more detail the reasons of a decrease in  PDR5  expression during prolonged incubation with the substances under study.

Fig. 5 Does this result indicate that, in addition to PDR5 other proteins are also involved  the interaction of cells with both compounds.

Author Response

Lines  84-85 It is unclear how the data on the sensitivity of yeast to the test substances lead to the conclusion: «These results suggest that human pathogenicity and fluconazole resistance does not affect the synergistic antifungal activity of the drug combination.»

Lines  84-85: The phrase ‘human pathogenicity’, not inferred from the results, has been deleted.

Fig. 4. - It would be interesting to discuss in more detail the reasons of a decrease in  PDR5  expression during prolonged incubation with the substances under study.

Line 301-303: We added one speculation about a decrease in the expression to discussion section of the revised manuscript.

Fig. 5 Does this result indicate that, in addition to PDR5 other proteins are also involved the interaction of cells with both compounds.

Thank you for your feedback. But, on the contrary, the results shows that PDR5 probably depends on synergistic antifungal activity.